# Effect of Tomato Nutrient Complex on Blood Pressure: A Double Blind, Randomized Dose–Response Study

**DOI:** 10.3390/nu11050950

**Published:** 2019-04-26

**Authors:** Talia Wolak, Yoav Sharoni, Joseph Levy, Karin Linnewiel-Hermoni, David Stepensky, Esther Paran

**Affiliations:** 1Hypertension Unit, Soroka University Medical center and Ben-Gurion University of the Negev, Beer Sheva 84101, Israel; twolak@bgu.ac.il (T.W.); paran@bgu.ac.il (E.P.); 2Department of Clinical Biochemistry and Pharmacology, Ben-Gurion University of the Negev, Beer Sheva 84105, Israel; lyossi@bgu.ac.il (J.L.); davidst@bgu.ac.il (D.S.); 3Lycored Ltd., Secaucus, NJ 07094, USA; Karin.Hermoni@lycored.com

**Keywords:** hypertension, carotenoids, tomato extract, lycopene, phytoene, phytofluene, bioavailability

## Abstract

Oxidative stress is implicated in the pathogenesis of essential hypertension, a risk factor for cardiovascular morbidity and mortality. Tomato carotenoids such as lycopene and the colorless carotenoids phytoene and phytofluene induce the antioxidant defense mechanism. This double-blind, randomized, placebo-controlled study aimed to find effective doses of Tomato Nutrient Complex (TNC) to maintain normal blood pressure in untreated hypertensive individuals. The effect of TNC treatment (5, 15 and 30 mg lycopene) was compared with 15 mg of synthetic lycopene and a placebo over eight weeks. Results indicate that only TNC treatment standardized for 15 or 30 mg of lycopene was associated with significant reductions in mean systolic blood pressure (SBP). Treatment with the lower dose standardized for 5 mg of lycopene or treatment with 15 mg of synthetic lycopene as a standalone had no significant effect. To test carotenoid bioavailability, volunteers were treated for four weeks with TNC providing 2, 5 or 15 mg lycopene. The increase in blood levels of lycopene, phytoene, and phytofluene was dose dependent. Results suggest that only carotenoid levels achieved by the TNC dose of 15 mg lycopene or higher correlate to a beneficial effect on SBP in hypertensive subjects while lower doses and lycopene alone do not.

## 1. Introduction

Hypertension (HT) is a major risk factor for cardiovascular morbidity and mortality. Prior studies provide strong evidence that oxidative stress and inflammatory processes correlate with the pathogenesis of HT. Although many drugs related to this chronic pathology are available, the high profile of side effects of anti-HT drugs warrant alternative and complementary treatment for blood pressure (BP) control [1,2]. This includes lifestyle modifications, especially dietary interventions such as consumption of more fruits and vegetables [3,4]. In line with these, the antioxidant lycopene has gained considerable attention in improving vascular function and regulating BP [5]. However, intervention trials investigating the role of lycopene supplementation or lycopene-containing foods in BP regulation have produced conflicting results. Several studies demonstrate that oral supplementation with tomato extract or tomato juice significantly decreases BP [5,6,7,8,9], while other studies show no relation [10] or no obvious association [11,12]. Interestingly one study even shows that lycopene can elevate BP [13].

In a meta-analysis, Ried et al. [14] reviewed four studies investigating the effect of lycopene-containing foods on BP and concluded that lycopene treatment could effectively decrease systolic BP (SBP), but had no statistically significant effect on diastolic BP (DBP). In an updated meta-analysis of intervention trials, Li et al. [15] similarly concluded a significant reduction in SBP by lycopene, but no significant effect on DBP. An important finding in this report is that lycopene treatment is more efficient in decreasing higher SBP (above 120 mmHg), but has no effect on normal BP.

Tomato products or extracts are used as the lycopene source in most of these studies. The extracts contain lycopene, β-carotene and the colorless carotenoids phytoene and phytofluene, in addition to a myriad of other active nutrients such as tocopherols and polyphenols. Many of these are strong antioxidants and known inducers of the antioxidant defense pathway, which is one of the mechanisms for the cardiovascular protective effect. The fact that these preparations are not pure lycopene has been frequently ignored, for example, in meta-analysis publications [14,15]. Thus, one of the major aims of the current work was to compare the effect on BP of a pure synthetic lycopene to a proprietary tomato extract. To complement our previous results showing the improvement of BP by natural antioxidants from tomato extract in pre-HT patients [6,9], we tested its BP-lowering effects in Stage 1 and 2 HT patients, attempting to establish a dose dependency and to show bioavailability of the major carotenoids present in tomato extract.

## 2. Materials and Methods

### 2.1. Study Population

The original sample size was 26 subjects in each study arm (see Section 2.5). However, following completion of treatment of the first 46 subjects (9–10 subjects in each treatment arm), an interim analysis was performed. Based on this, the study recruitment was stopped, and all study data were analyzed.

Sixty-one hypertensive individuals (according to the 2017 ACC guidelines) [16] without anti-HT treatment, aged 35–60 years, were recruited from primary-care clinics and through advertisements posted in local newspapers. The recruited patients had BP values in the range of 130 < SBP < 145 mmHg or 80 < DBP < 95 mmHg, measured as detailed in Section 2.3. Subjects treated for HT or dyslipidemia; who had any suspected allergy to tomato, carotenoids or α-tocopherol; or were taking vitamins or other food additives were excluded from the study. Smokers, persons with diabetes or cardiovascular, gastrointestinal, hepatic or malignant diseases were also excluded. All included subjects signed an informed consent form presented to them by the researchers after receiving an explanation regarding the course of the trial. The local Helsinki ethics committee approved the study protocol (approval No. 4594, 2008). The study was registered at ClinicalTrials.gov with the Identifier: NCT00637858.

### 2.2. Materials

Tomato Nutrient Complex, a proprietary tomato extract, was supplied by Lycored Ltd., Beer Sheva, Israel at doses corresponding to 5, 15 and 30 mg lycopene. Lycored also supplied identical-looking capsules with 15 mg synthetic lycopene (18–20% cis isomers) and placebo capsules containing soybean oil. The TNC 15-mg capsule contained the indicated amounts of the following tomato phytonutrients: lycopene (6%; 8–10% cis isomers), phytoene (1%), phytofluene (1%); β-carotene (0.15–0.2%), vitamin E (2%), and other fat-soluble phytonutrients naturally present in the tomato, suspended in tomato oleoresin oil.

### 2.3. Procedure

At the screening visit, held at the Hypertension Outpatient Clinic of the Soroka University Medical Center, a thorough physical examination was performed, and a comprehensive medical and dietary history was collected for each participant. Blood pressure, pulse rate, height and weight were measured, and body mass index (BMI) was calculated. BP was measured after 10 min of rest in a sitting position using a new and calibrated Omron HEM–705CP electronic semiautomatic sphygmomanometer (Tokyo, Japan), which was used only for this study. Recorded BP was calculated as the average of three serial measurements if the difference between them was <8 mmHg for SBP and <5 mmHg for DBP. A trained research nurse blinded to the study periods and treatment took all measurements. BP measurements were taken at the same hour of the morning after abstinence from food and caffeine for a minimum of 30 min.

Allocated participants started treatment with a four-week single-blind placebo run-in period, in which they were asked to consume one placebo capsule a day. This phase was used to confirm that participants indeed had HT. Only subjects with BP values of 130 < SBP < 145 mmHg or 80 < DBP < 95 mmHg were randomized for the double-blind placebo-controlled treatment phase. The study was performed under complete blind conditions (single-blind or double-blind, in accordance with the study phase). Knowledge of the randomization list was limited to the independent study statistician responsible for the creation of the list (Dr. Michael Friger, Department of Epidemiology, Ben-Gurion University) and the LycoRed designee responsible for preparation of the study medication. The subjects were allocated sequential subject numbers, and the investigator entered the subject randomization number into each subject’s file. There were five treatment arms: TNC 5 mg, TNC 15 mg, TNC 30 mg, synthetic lycopene (15 mg) and a placebo administered once daily for eight weeks. Participants were blinded to the different study periods and were instructed to take the capsules with the main meal of the day to improve absorption of its ingredients. No other dietary supplements were allowed throughout the study, and participants were instructed to keep their usual dietary habits. Follow-up visits were held every two weeks and included a short clinical evaluation, and BP and pulse rate measurements. Study medications were dispensed, and compliance was verified by counting the remaining capsules and giving reinforcement at each visit.

Blood was drawn after an overnight fast at the end of the run-in phase and the end of the eight-week treatment phase. Blood analysis for safety included: complete blood count; plasma levels of glucose, urea, uric acid, sodium, potassium, creatinine, aspartate amino transferase (AST), alanine amino transferase (ALT), alkaline phosphatase (ALP), lactate dehydrogenase (LDH) and bilirubin; and blood lipid profile: cholesterol, high-density lipoprotein (HDL), low-density lipoprotein (LDL) and triglycerides. Additional plasma samples were stored at −70 °C. Stored samples were extracted with ethanol and hexane/dichloromethane, and then analyzed for lycopene using a high-performance liquid chromatography (HPLC)-based analytical method.

### 2.4. Steady-State Bioavailability of Tomato Carotenoids TNC Dose–Response

The study was performed with a different group of subjects from the ones participating in the BP study. It was conducted at the Endocrine Laboratory, Department of Clinical Biochemistry and Pharmacology of the Ben-Gurion University and the Soroka University Medical Center and was approved by the Helsinki Committee of the medical center (approval No. 0012-15-SOR). Twenty-five healthy volunteers, aged 20–40, were recruited from students of Ben-Gurion University through advertisements posted on the university electronic bulletin board. Written informed consent was obtained from all volunteers prior to entry into the double-blind, randomized, cross-over study. In each experiment, baseline blood samples were collected after overnight fasting, and treatment with TNC at 2, 5 or 15 mg of lycopene was begun for four weeks, followed by a four-week washout period between experiments without consumption of any supplement. The capsules with the different doses were identical in appearance, and the different doses were randomly distributed to about one-third of the volunteers in each experiment. Both the volunteers and the researchers were blinded to the received doses. Participants were instructed to consume one capsule per day with their main meal and to keep their usual dietary habits throughout the experimental and washout periods. Blood samples were taken after two, three and four weeks of treatment, after overnight fasting and 24 h after taking the last supplement. Compliance was verified by counting the remaining capsules and it was 90% or higher.

Carotenoid analysis was performed after ethanol and hexane/dichloromethane (4/1) extraction by dedicated HPLC method using a C30 reverse phase column, gradient elution and detection at multiple wavelengths using a photodiode array detector.

### 2.5. Statistical Methods

Sample size calculations were performed by using appropriate formulas based on 80% power and a two-sided α = 0.05 with assumption of a standard deviation of DBP equal to 3.8 mmHg. A clinically significant difference in DBP was determined at 3.0 mmHg. For this determination, the sample size was 26 patients in each treatment arm—in all, 130 participants.

The paired t-test or non-parametric signed-rank test was applied for testing the differences of the continuous assessments between all visits to the baseline. An ANOVA model using the Duncan method was applied for testing the differences in blood pressure changes between all study groups. All tests applied were two-tailed, and a *p*-value of 5% or less was considered statistically significant. The data were analyzed using the SAS^®^ version 9.1 for Windows (SAS Institute, Cary, North Carolina). The missing data for early withdrawals who were not replaced and who attended at least four weeks of the double-blind placebo-controlled phase of the study were handled as LOCFs (last observation carried forward).

For the bioavailability study, summary statistics were calculated using a GraphPad Prism 5.0 program. Paired t-tests were used to compare the carotenoid concentrations at different time points. A *p*-value of <0.05 was deemed statistically significant.

## 3. Results

### 3.1. Demographics and Baseline Characteristics

Sixty-one patients with BP values in the range of 130 < SBP < 145 mmHg or 80 < DBP < 95 mmHg were enrolled in the study and began treatment with a four-week single-blind placebo run-in phase. These patients were randomized for the double-blind placebo-controlled treatment phase. At enrollment, there were 12 subjects in each of the following arms: TNC 5 mg, TNC 15 mg, synthetic lycopene 15 mg and placebo, in addition to 13 subjects in the TNC 30-mg arm. Forty-six subjects completed the eight-week treatment period and 15 (3 in each arm) dropped out of the study prematurely. There were neither adverse effects reported during the entire study period nor any significant changes in glucose, urea, creatinine, uric acid, sodium, potassium, chloride, cholesterol, triglycerides, AST, ALT, ALP, LDH, HDL, LDL levels, or in the blood count parameters.

The treatment arms were comparable with respect to all demographic and baseline characteristics (Table 1). Baseline SBP and DBP measurements were not statistically significantly different (Table 1). The mean age was 52.4 ± 8.2 years, and 73.8% were male. The mean SBP and DBP were 135.2 ± 7.4 mmHg and 82.0 ± 11.7 mmHg, respectively. There were no statistically significant differences between the five arms in the baseline plasma lycopene concentrations; however, the mean concentration of the TNC 15-mg arm was somewhat higher than the other arms.

### 3.2. Changes in BP Values during the 8-Week Double-Blind Placebo-Controlled Treatment Phase

The average SBP in the five treatment arms during eight weeks of treatment is presented in Table 2. The change in SBP in the treatment arms during this same period are presented in Figure 1. Treatment with TNC containing 15 or 30 mg lycopene was associated with statistically significant reductions in mean SBP at almost all time points from two to eight weeks. Similar effects were not observed following treatment with 5 mg of TNC, 15 mg of synthetic lycopene or the placebo. A comparison between treatment arms revealed that the reduction in SBP in the TNC 15-mg arm following eight weeks of treatment was statistically different from that of the TNC 5-mg arm, the placebo arm and the synthetic lycopene arm (*p* = 0.0495). Remarkably, the rate of reduction in SBP with time of treatment for the 15-mg and 30-mg arms is almost parallel (Figure 1), suggesting that TNC containing 15 mg lycopene is both necessary and sufficient to normalize SBP.

The average DBP in the five treatment arms during the eight-week treatment is presented in Table 3. The changes in DBP in the five arms during this period are presented in Table 4. A significant reduction in DBP from baseline to eight weeks was observed only in the TNC 15-mg arm (−4.1 ± 5.0, *p* = 0.038, Table 4). There were no significant differences in DBP when treatment arms were compared.

### 3.3. Changes in BP Values during the Single-Blind 16-Week Study Extension

Of the 46 subjects who completed the first phase of eight weeks of treatment, 31 agreed to continue for a single-blind four-month extension study with TNC containing 15 mg of lycopene. Four subjects ended this phase prematurely. The average SBP and DBP of the 27 subjects who completed the extension phase were compared to the average baseline values of the same subjects, measured at the randomization stage before the eight weeks of treatment. The results of SBP and DBP are presented in Table 5 and Figure 2. A statistically significant reduction in mean SBP was evident at all time points from one to four months. This result suggests that the SBP-reducing effect of TNC is long lasting. The changes in DBP from baseline to Month 4 were not significant.

### 3.4. Changes from Baseline of Plasma Levels of Lycopene

There was large variability in plasma lycopene baseline values (Table 6), and in the changes from baseline to eight weeks of treatment (Figure 3). Due to this variability and the small number of tested samples, there was no significant change in lycopene concentration in any of the arms, but the highest change was observed in the synthetic lycopene arm (Figure 3).

### 3.5. Bioavailability of Tomato Carotenoids during 4 Weeks of Daily TNC Supplementation—A Dose– Response Study

Because of the large variability in the results of carotenoid plasma concentrations in the BP study patients (Section 3.4) and the low number of available samples, we conducted a separate bioavailability study of TNC with a different group of volunteers to evaluate the dose–response of steady-state carotenoid plasma concentrations during daily TNC treatment. A statistically significant increase in plasma lycopene was evident from Week 2 for the three supplemented doses (Table 7). For phytofluene, such an increase was not detected at all time points for the lowest dose (TNC 2 mg), but was evident at the highest dose (TNC 15 mg), and at Weeks 3 and 4 with TNC 5 mg. For phytoene, a significant increase was already detected for the two higher doses at Week 2 and at the lowest dose only at Week 4. No significant change was detected in the plasma concentrations of β-carotene, a control carotenoid that is found in very low amounts in the TNC preparations (e.g., 0.8 mg in the TNC 15 mg).

To calculate the average net increase in carotenoid concentrations, individual baseline concentrations were subtracted from values at other time points (Figure 4). The increase in plasma lycopene and phytofluene concentrations were significantly lower with TNC 5 mg and TNC 2 mg than with TNC 15 mg. This result suggests that the lower increase in these carotenoid concentrations for TNC 5 mg as compared to TNC 15 mg was not enough to drive the reduction in SBP in the HT subjects (Table 3). For phytoene, there was a significant difference between TNC 15 and TNC 2 mg, but not TNC 5 mg. When comparing the changes in carotenoid concentrations between TNC 5 mg and TNC 2 mg, a significantly higher increase was evident at the three time points only for phytoene concentrations, but not for lycopene and phytofluene.

A small continuous increase in the three carotenoid concentrations was observed from Week 2, to Weeks 3 and 4. However, there were no statistically significant differences between the concentrations at these time points, which suggests that steady-state concentrations were already attained after two weeks of TNC supplementation.

## 4. Discussion

The aims of the current study were to perform a dose–response analysis and uncover the optimal effective dose of a proprietary Tomato Nutrient Complex supplement in maintaining blood pressure within a normal range among untreated HT individuals and to compare the effect of TNC to that of synthetic lycopene. The bioavailability of the major tomato carotenoids was studied in a separate group of volunteers to gain insight into their relative contribution to the BP lowering effect of TNC.

Our study has several limitations. The minimum time between food and caffeine intake was determined to be 30 min. However, this time may be too short to prevent any effect of food intake on the BP measurement. In addition, the exact time and the type of food/beverages taken were not recorded; thus, it was not possible to estimate the effect of the food on BP analysis. The study lacks quantitative information on consumption of fruit and vegetables at baseline and at follow-up times. However, the lycopene plasma concentrations at baseline, which partially reflect this consumption, were not statistically different between study arms (Table 1), suggesting that this did not have a major effect on the study results. Although the participants were encouraged to maintain their dietary habits, there is no quantitative information on their food consumption during the study period to confirm that they did. In addition, there is no information on BMI or physical activity before and during the study, factors that may affect BP. These limitations should be considered in future studies, but it should be emphasized that, despite these limitations, the study resulted in showing significant effects of TNC on SBP.

Results of this double-blind, randomized, placebo-controlled study indicate that treatment with TNC standardized to contain 15 or 30 mg of lycopene was associated with statistically significant reductions in mean SBP. Treatment with the lower dose of TNC standardized for 5 mg of lycopene or treatment with 15 mg of synthetic lycopene did not produce a significant effect. DBP was not significantly different from the baseline with any treatment or control arm. Subjects from all treatment arms participated in an additional single-blind extension period for four months with TNC containing 15 mg lycopene, which resulted in a significant reduction of SBP that was persistent throughout the extension period. Thus, the reduction of SBP appears to be of a long-term nature. During the study and extension period, all treatments were safe and well tolerated, and no treatment-related adverse effects were reported.

The current results are in line with the recent meta-analysis of the effects of tomato products on BP [14,15] demonstrating a significant reduction of SBP, but no statistically significant effect on DBP. Subgroup analyses within the six studies [1,6,7,8,9,13], which met the inclusion criteria in the more recent study [15], demonstrated that higher dosage of tomato-derived supplements (containing more than 12 mg lycopene per day) could significantly lower SBP, whereas lower doses did not show a significant effect. This is similar to the results of the current study, which did not indicate a significant SBP reduction with TNC corresponding to 5 mg lycopene per day, but did show an effect at 15 and 30 mg. The low dose is probably the reason for the negative results in Paterson et al.’s study [13], which used only 4.5 mg/day lycopene, an amount close to the non-effective lycopene dose in the current study. Another important factor obtained in the meta-analysis was related to the baseline SBP. A significant reduction of SBP was observed if participants had higher baseline SBP (SBP ≥ 120 mmHg). Although this issue was not examined in the current study, it is important to note that the baseline SBP of all participants was above the threshold level (120 mmHg) suggested by the meta-analysis.

Remarkably, treatment with 15 mg of synthetic lycopene as a standalone did not cause a significant reduction in SBP. Apparently, this was not due to lower bioavailability of lycopene in the synthetic preparation. Actually, the mean increase in blood lycopene between the baseline and eight weeks was higher with the synthetic lycopene as compared to treatment with TNC containing 15 and 30 mg lycopene. Synthetic lycopene is different from tomato-derived lycopene in the ratio of cis:trans isomers—about 20% cis-isomers are in the synthetic lycopene and 10% in TNC. However, this difference cannot explain the better efficiency of TNC in lowering SBP, as we showed previously that the cis:trans ratio in supplemented lycopene does not affect the ratio in plasma and tissues [17]. The effects of purified lycopene was tested in another study on biomarkers of oxidative stress [18]. In that study, healthy volunteers were treated with 0, 6.5, 15 or 30 mg lycopene/day for eight weeks, which significantly increased plasma lycopene levels in agreement with our results. Interestingly, it was found that pure lycopene supplementation at all doses did not affect biomarkers of lipid peroxidation, whereas biomarkers of DNA damage were affected only at the highest dose of 30 mg/day of pure lycopene. This is similar to the lack of effect of 15 mg pure lycopene on BP in the current study.

One explanation for the lower efficacy of the synthetic lycopene is the presence of other active nutrients in the tomato extract preparation. Indeed, in a previous study [19], we found that the anti-cancer effects of carotenoids and other phytonutrients present in tomato extract (e.g., lycopene, phytoene and phytofluene) resides in their combined activity, which is synergistically higher than the activity of each compound alone. To assess the potential contribution of other tomato constituents to the BP-lowering effect, we analyzed the bioavailability of these carotenoids following treatment with different doses of TNC in a group of 25 healthy volunteers. Although the characteristics of these volunteers are different from those of the participants of the BP study, the information about the differences in the dose–response of the three major tomato carotenoids can shed light on the results of the BP study, as discussed below. The increase in blood levels of lycopene, phytoene and phytofluene was dose dependent, and was significantly higher with TNC 15 mg as compared to TNC 5 mg. A large increase in plasma concentrations of lycopene, phytofluene, and phytoene was already evident after two weeks of treatment. This correlates well with the reduction of SBP with TNC 15 mg, which was statistically significant after two weeks of treatment, suggesting that the increase in these tomato constituents was the reason and the driving force for the SBP reduction. It should be noted that the changes in phytofluene and phytoene plasma concentrations between the non-effective dose of TNC 5 mg and the effective TNC 15 mg was more pronounced than that of lycopene, which suggests that these carotenoids (and possibly other components of TNC) have an important role in reducing SBP. Such a role for phytofluene and phytoene can also explain the lower effect of pure synthetic lycopene, which does not contain these carotenoids or any other tomato component. This option is supported by a study showing that a reduction in UV-induced erythema by tomato extract enriched with phytofluene and phytoene was larger than that achieved with non-enriched tomato extract, whereas the lowest erythema reduction was found with synthetic lycopene [20].

An important question not addressed in the current study is what the mechanism might be by which the supplemented tomato carotenoids reduce BP. A possible answer is suggested by a study in endothelial cells [21], in which Di Tumo et al. found that lycopene and β-carotene reduce TNFα-induced inflammation in endothelial cells in culture. This effect was associated with reduced reactive oxygen species and nitrotyrosine [an index of interaction of NO with superoxide anion (O_2_^−^)] and an increase in NO and cGMP, which are known to cause vascular relaxation. The antioxidant effect of the carotenoids, which led to the observed effects in the endothelial cells, and possibly to lower SBP in the current human study, probably resulted from activation of the antioxidant response element and the Nrf2 transcription system that is induced by carotenoids, as we reported previously [22,23].

## 5. Conclusions

TNC containing 15 mg and 30 mg lycopene was well tolerated and showed efficacy in reducing SBP in the HT population, while lower doses and standalone pure lycopene were not sufficient to induce similar effects. The content of lycopene in raw tomatoes is 2.5–4 mg per 100 g [24]. Thus, reasonable consumption of tomatoes, e.g., 100–200 g per day, will supply 4–8 mg lycopene, but not enough to drive SBP reduction; therefore, inclusion in the daily diet of other lycopene-rich foods such as tomato products and supplements is recommended. It is not clear whether a supplement such as TNC can be used as a standalone treatment or in conjunction with other treatments. However, our previous study that used tomato extract in treated but uncontrolled hypertensive patients [9] suggests that patients treated with various antihypertensive drugs can benefit from the addition of TNC.

## Figures and Tables

**Figure 1 nutrients-11-00950-f001:**
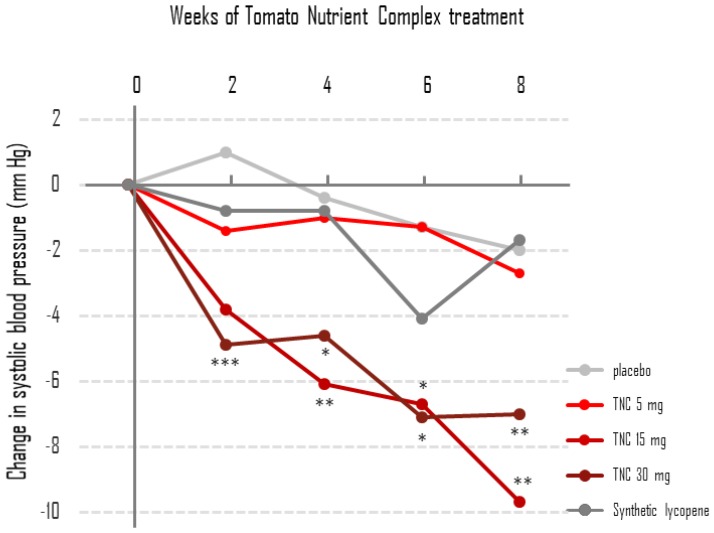
Tomato Nutrient Complex containing 15 mg and 30 mg lycopene reduces systolic blood pressure. Results are the mean of the individual changes from baseline to each time point from two to eight weeks. SDs are not shown for graph simplicity. *N* is the same as in Table 2. * *p* < 0.05; ** *p* < 0.01; *** *p* < 0.001 for changes to baseline within treatment arm.

**Figure 2 nutrients-11-00950-f002:**
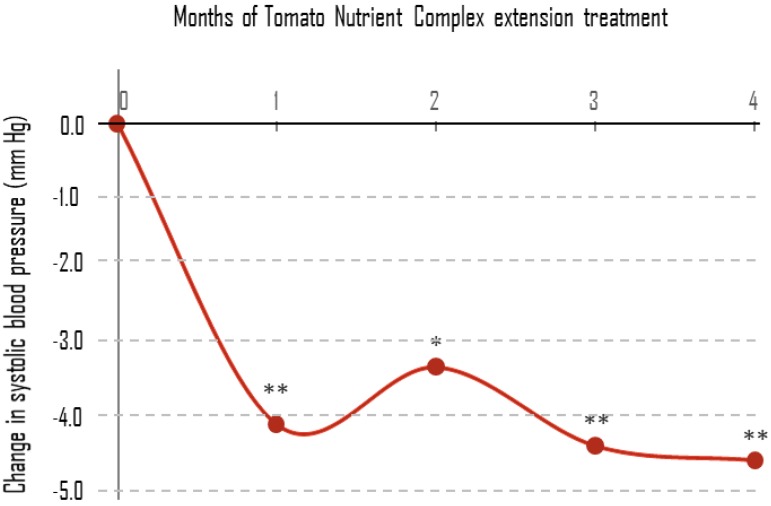
Changes in SBP values during the single-blind 16-week study extension. SBP values are the mean for the 27 subjects who completed this phase. * *p* < 0.05; ** *p* < 0.01 for changes to baseline within treatment arm.

**Figure 3 nutrients-11-00950-f003:**
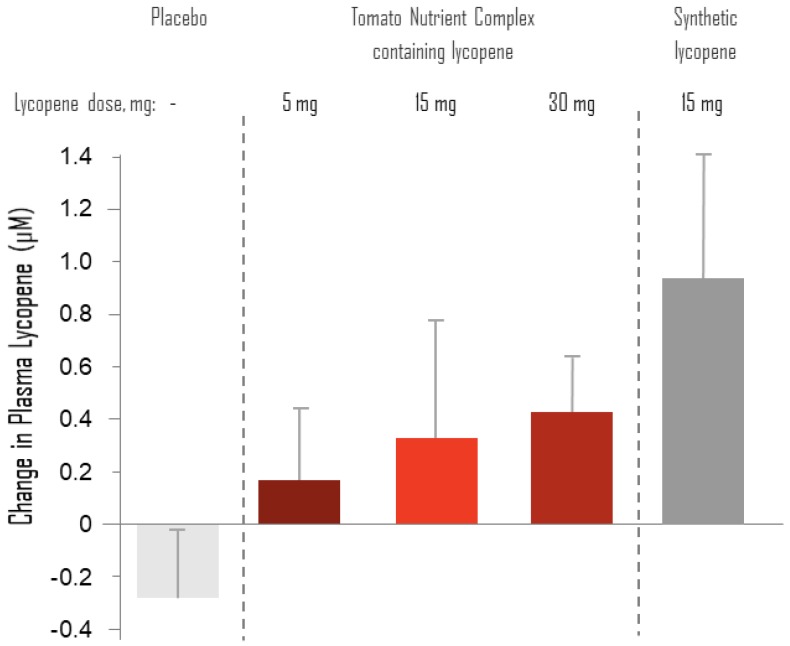
Changes of plasma levels of lycopene from baseline to eight weeks. Results are the mean ± SD of the changes from baseline to Week 8 within treatment arms. *N* was 9, 7, 7, 9, and 7 for the placebo, TNC 5 mg, TNC 15 mg, TNC 30 mg and synthetic lycopene arms, respectively.

**Figure 4 nutrients-11-00950-f004:**
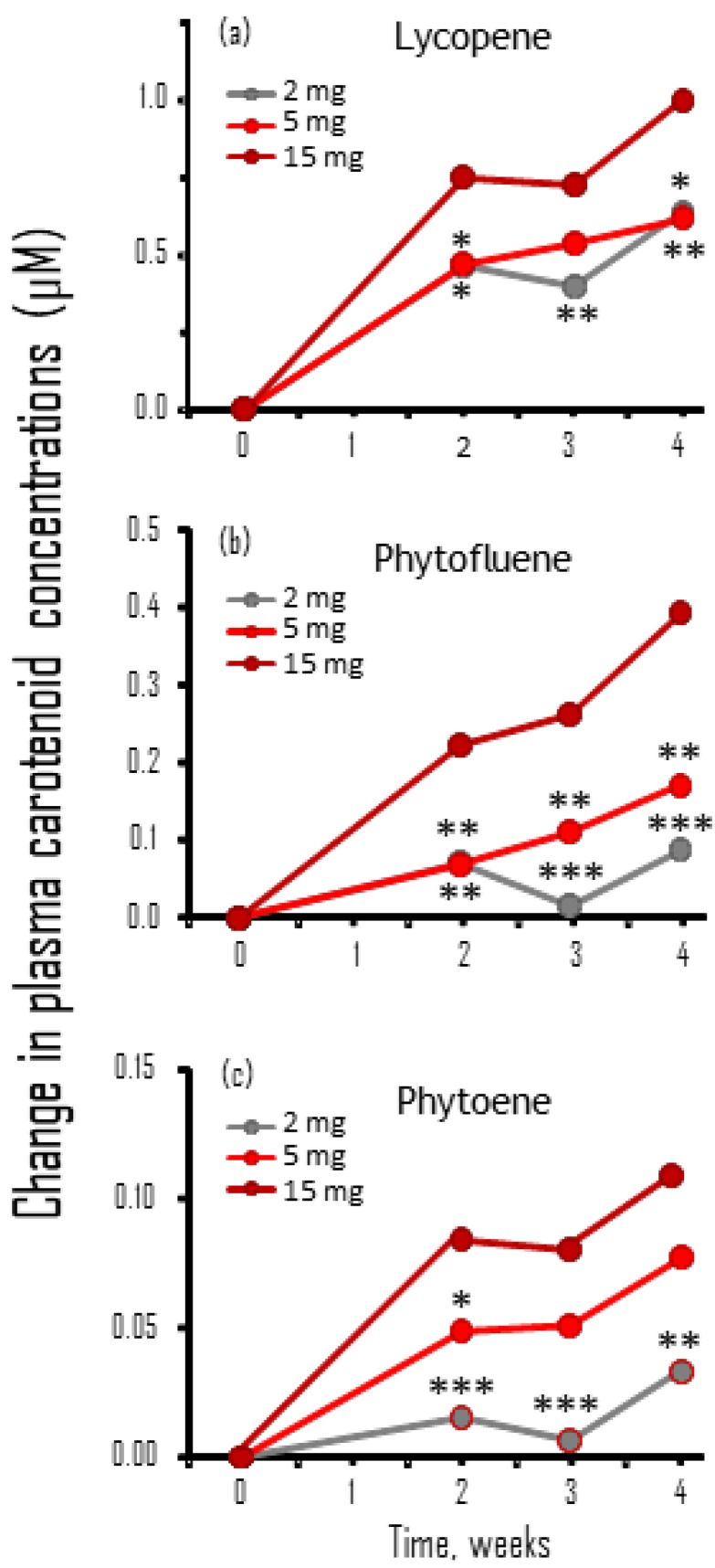
The increase in lycopene, phytofluene and phytoene concentrations after 2–4 weeks of treatment with TNC were significantly lower with TNC 5 mg and TNC 2 mg than with TNC 15 mg: (**a**) lycopene; (**b**) phytofluene; and (**c**) phytoene. Please note that the scale in the Y-axis is different for the three carotenoids. Results are the mean of individual changes in carotenoid concentrations of 25 subjects (TNC 2 mg and 15 mg) or 24 subjects (TNC 5 mg). SDs are not shown for graph simplicity. * *p* < 0.05; ** *p* < 0.01; *** *p* < 0.001 for the difference at each time point between TNC 15 mg to TNC 2 mg or TNC 5 mg.

**Table 1 nutrients-11-00950-t001:** Baseline characteristics of the study population.

Parameter	TNC 5 mg	TNC 15 mg	TNC 30 mg	Synthetic Lycopene 15 mg	Placebo	All
Age (years) ^1^	53.5 ± 8.0 (12)	54.0 ± 8.6 (12)	51.0 ± 10.1 (13)	52.1 ± 8.4 (12)	51.8 ± 6.3 (12)	52.4 ± 8.2 (61)
Males (%)	66.7% (8)	75.0% (9)	69.2% (9)	66.7% (8)	83.3% (10)	72.1% (44)
Females (%)	33.3% (4)	25.0% (3)	30.8 (4)	33.3% (4)	16.7% (2)	27.9% (17)
SBP (mmHg) ^1^	133.6 ± 7.8 (12)	137.4 ± 5.6 (12)	136.4 ± 7.8 (13)	132.8 ± 9.3 (12)	135.7 ± 6.0 (12)	135.2 ± 7.4 (61)
DBP (mmHg) ^1^	82.9 ± 9.3 (12)	83.8 ± 6.3 (12)	77.5 ± 21 (13)	82.3 ± 6.1 (12)	83.7 ± 8.7 (12)	82.0 ± 11.7 (61)
Plasma lycopene (µM)	0.97 ± 0.49 (7)	1.55 ± 0.86 (7)	0.80 ± 0.74 (10)	0.91 ± 0.95 (8)	1.01 ± 1.24 (10)	0.93 ± 0.68 (42)
TG (mg/dL)	152.0 ± 70.4 (11)	164.3 ± 65.3 (10)	124.5 ± 75.3 (12)	115.1 ± 50.5 (10)	166.0 ± 103.1 (12)	
HDL chol (mg/dL)	52.0 ± 12.2 (10)	49.5 ± 10.1 (10)	54.4 ± 10.0 (12)	59.1 ± 18.2 (10)	52.1 ± 11.7 (11)	
LDL chol (mg/dL)	142.1 ± 34.6 (10)	107.4 ± 26.0 (9)	104.8 ± 31.1 (11)	134.2 ± 29.5 (10)	112.9 ± 30.1 (9)	

^1^ Age, BP, lycopene, TG, HDL, and LDL values are mean ± SD (*N*).

**Table 2 nutrients-11-00950-t002:** SBP during eight weeks of treatment.

SBP ^1^ (mmHg)	TNC 5 mg	TNC 15 mg	TNC 30 mg	Synthetic Lycopene 15 mg	Placebo
Baseline	133.6 ± 7.8 (12)	137.4 ± 5.6 (12)	136.4 ± 7.8 (13)	132.8 ± 9.3 (12)	135.7 ± 6.0 (12)
Week 2	132.7 ± 8.9 (10)	133.7 ± 5.3 (12)	131.5 ± 8.0 (13)	133.2 ± 8.9 (11)	137.1 ± 10.3 (11)
Week 4	133.1 ± 9.2 (9)	130.7 ± 4.7 (10)	131.7 ± 10.5 (11)	133.4 ± 7.2 (9)	134.5 ± 8.7 (10)
Week 6	132.8 ± 9.2 (9)	130.2 ± 10.6 (9)	129.9 ± 13.9 (10)	130.1 ± 9.8 (9)	133.7 ± 9.9 (9)
Week 8	131.4 ± 7.7 (9)	127.2 ± 6.3 (9)	130.0 ± 13.0 (10)	132.6 ± 5.1 (9)	133.0 ± 6.7 (9)

^1^ SBP values are mean ± SD (*N*).

**Table 3 nutrients-11-00950-t003:** DBP during eight weeks of treatment.

DBP ^1^ (mmHg)	TNC 5 mg	TNC 15 mg	TNC 30 mg	Synthetic Lycopene 15 mg	Placebo
Baseline	82.9 ± 9.3 (12)	83.8 ± 6.3 (12)	77.5 ± 21 (13)	82.3 ± 6.1 (12)	83.7 ± 8.7 (12)
Week 2	81.4 ± 8.2 (10)	80.7 ± 7.1 (12)	80.5 ± 7.9 (13)	84.5 ± 6.1 (11)	84.2 ± 8.2 (11)
Week 4	82.8 ± 9.9 (9)	81.3 ± 3.9 (10)	82.6 ± 9.1 (11)	84.9 ± 6.6 (9)	83.0 ± 8.8 (10)
Week 6	82.6 ± 6.4 (9)	78.8 ± 7.5 (9)	72.3 ± 23 (10)	81.6 ± 6.0 (9)	83.2 ± 7.9 (9)
Week 8	81.7 ± 7.9 (9)	78.6 ± 7.9 (9)	74.4 ± 23 (10)	84.6 ± 5.0 (9)	85.3 ± 7.0 (9)

^1^ DBP values are mean ± SD (*N*).

**Table 4 nutrients-11-00950-t004:** Change in DBP from baseline during eight weeks of treatment.

DBP (mmHg)	TNC 5 mg	TNC 15 mg	TNC 30 mg	Synthetic Lycopene 15 mg	Placebo
	Mean ± SD (*p* ^1^)	Mean ± SD (*p*)	Mean ± SD (*p*)	Mean ± SD (*p*)	Mean ± SD (*p*)
Week 2	−2.9 ± 7.7 (0.265)	−3.2 ± 4.5 (0.032)	3.0 ± 4.9 (0.672)	0.9 ± 2.9 (0.331)	0.0 ± 5.0 (1.000)
Week 4	−0.9 ± 5.1 (0.613)	−1.9 ± 5.1 (0.268)	6.8 ± 7.0 (0.422)	1.2 ± 4.6 (0.449)	−0.4 ± 8.6 (0.887)
Week 6	−1.1 ± 5.8 (0.582)	−3.9 ± 6.5 (0.112)	−3.5 ± 5.9 (0.092)	−2.1 ± 4.5 (0.198)	−1.4 ± 9.8 (0.671)
Week 8	−2.0 ± 6.3 (0.371)	−4.1 ± 5.0 (0.038)	−1.4 ± 5.7 (0.454)	0.9 ± 5.6 (0.645)	0.7 ± 5.8 (0.740)

^1^*p*-value for changes to baseline within treatment arms. *N* is the same as in Table 3.

**Table 5 nutrients-11-00950-t005:** Mean SBP and DBP and changes from baseline in BP values during four-month study extension.

	SBP (mmHg) ^1^	Change of SBP ^1^ from Baseline (mmHg)	*p*-Value for Change	DBP (mmHg) ^1^	Change of DBP ^1^ from Baseline (mmHg)	*p*-Value for Change
Baseline	135.1 ± 8.76			85.1 ± 6.6		
Month 1	130.9 ± 8.6	−4.2 ± 6.5	0.0024	84.3 ± 6.2	−0.9 ± 5.7	0.422
Month 2	131.7 ± 10	−3.5 ± 7.4	0.0211	81.6 ± 15.5	−3.6 ± 15.8	0.252
Month 3	130.6 ± 8.8	−4.5 ± 7.0	0.0024	82.7 ± 7.7	−2.4 ± 6.3	0.056
Month 4	130.4 ± 10.9	−4.7 ±6.7	0.0012	83.9 ± 7.3	−1.4 ± 4.3	0.362

^1^ BP values and changes in BP values are mean ± SD for the 27 subjects who completed the four-month extension phase.

**Table 6 nutrients-11-00950-t006:** Plasma lycopene value at baseline and at eight weeks of treatment.

SBP ^1^ (mmHg)	TNC 5 mg	TNC 15 mg	TNC 30 mg	Synthetic Lycopene 15 mg	Placebo
Baseline	0.97 ± 0.49 (7)	1.55 ± 0.86 (7)	0.80 ± 0.74 (9)	0.91 ± 0.95 (7)	1.01 ± 1.24 (9)
Week 8	1.13 ± 0.97 (7)	1.89 ± 1.22 (7)	1.23 ±1.07 (9)	1.84 ± 1.14 (7)	0.72 ± 0.72 (9)

^1^ Plasma lycopene values are mean ± SD (*N*).

**Table 7 nutrients-11-00950-t007:** Carotenoid plasma concentrations during 4-week treatment with TNC containing 2, 5 and 15 mg lycopene.

Carotenoid Plasma Concentration (µM)	Baseline Mean ± SD	Week 2 Mean ± SD (*p*) ^1^	Week 3 Mean ± SD (*p*) ^1^	Week 4 Mean ± SD (*p*) ^1^
**Lycopene**				
TNC 2 mg ^2^	1.021 ± 0.437	1.483 ± 0.481 (0.001)	1.354 ± 0.408 (0.009)	1.656 ± 0.537 (<0.000x)
TNC 5 mg ^2^	1.149 ± 0.322	1.629 ± 0.444 (<0.000x)	1.677 ± 0.470 (<0.000x)	1.767 ± 0.473 (<0.000x)
TNC 15 mg ^2^	1.036 ± 0.398	1.794 ± 0.453 (<0.000x)	1.754 ± 0.449 (<0.000x)	2.008 ± 0.627 (<0.000x)
**Phytofluene**				
TNC 2 mg ^2^	0.310 ± 0.147	0.382 ± 0.211 (0.171)	0.300 ± 0.126 (0.805)	0.398 ± 0.242 (0.128)
TNC 5 mg ^2^	0.354 ± 0.130	0.429 ± 0.190 (0.120)	0.456 ± 0.201 (0.046)	0.530 ± 0.263 (0.005)
TNC 15 mg ^2^	0.270 ± 0.111	0.503 ± 0.235 (<0.000x)	0.536 ± 0.228 (<0.000x)	0.650 ± 0.275 (<0.000x)
**Phytoene**				
TNC 2 mg ^2^	0.057 ± 0.040	0.073 ± 0.060 (0.300)	0.059 ± 0.037 (0.868)	0.091 ± 0.061 (0.028)
TNC 5 mg ^2^	0.046 ± 0.032	0.096 ± 0.050 (<0.000x)	0.092 ± 0.057 (0.001)	0.124 ± 0.058 (<0.000x)
TNC 15 mg ^2^	0.048 ± 0.031	0.133 ± 0.062 (<0.000x)	0.127 ± 0.068 (<0.000x)	0.158 ± 0.091 (<0.000x)
**β-carotene**				
TNC 2 mg ^2^	0.842 ± 0.391	0.975 ± 0.484 (0.289)	0.872 ± 0.486 (0.811)	0.998 ± 0.502 (0.224)
TNC 5 mg ^2^	0.947 ± 0.598	0.954 ± 0.580 (0.965)	0.919 ± 0.473 (0.866)	1.037 ± 0.533 (0.590)
TNC 15 mg ^2^	0.885 ± 0.534	0.968 ± 0.609 (0.618)	0.970 ± 0.553 (0.587)	1.057 ± 0.631 (0.315)

^1^*p*-value for difference from baseline within the same dose arm. ^2^*N* = 25 for TNC 2 mg and 15 mg. *N* = 24 for TNC 5 mg.

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
