# Peer review of "Effect of Tomato Nutrient Complex on Blood Pressure: A Double Blind, Randomized Dose–Response Study"

_nutrients, 2019, doi:10.3390/nu11050950_

Round 1
Reviewer 1 Report
The study by Wolak et al examines the contribution of a tomato nutrient complex (TNC) on blood pressure over an 8 week period. Overall the study is well designed, and was performed to understand the contribution (dose-dependent) of TNC, and how this compares to synthetic lycopene. The authors determine that TNC (>15 mg) is sufficient to suppress blood pressure over the study period.
In relation to the study participants:
- Is the consumption of fruits/vegetables different amongst the groups at baseline and at follow-up time periods? This is an important variable to assess given it is a significant contributor to blood/plasma carotenoid levels.
- CBC data was collected from participants at enrollment and followup, no note in made in the results as to whether this changed or remained constant, please comment.
Results:
- Table 1 should be updated to include demographic and clinical data associated with risk of hypertension BMI, race, ethnicity, HDL, LDL, smoking status, alcohol consumption and TG
- Data from Table 2 and Figure 2 suggest the TNC dose with the greatest benefit on blood pressure is 15 mg, however data in Figure 3 would suggest that serum increases in lycopene do not differ greatly between the different treatment doses. This is further supported by data from Table 6 which shows that lycopene dose not increase in serum in a dose-dependent nature.
Rather, phytofluene and phytoene increase in a dose-dependent nature - is there any evidence from the authors (or others) that these compounds may be having a beneficial effect on blood pressure?
- Can the authors provide insight into the proposed mechanism of action of THC and reductions in blood pressure within the discussion - given they comment it is not due to reductions in ROS production.
- The population used in this study has SBP/DBP values considered an "at risk" population, and THC reduced blood pressure to the normal range. Clinically, is the THC product anticipated to be used in conjunction with optimal medical therapy for the lowering of hypertension, or as a stand alone product? Or a preventative measure?
Table 6
p values should not be reported as 0.000, unless it truly is, these should be updated to P<0.000x.
Author Response
Review 1
The study by Wolak et al examines the contribution of a tomato nutrient complex (TNC) on blood pressure over an 8 week period. Overall the study is well designed, and was performed to understand the contribution (dose-dependent) of TNC, and how this compares to synthetic lycopene. The authors determine that TNC (>15 mg) is sufficient to suppress blood pressure over the study period.
In relation to the study participants:
- Is the consumption of fruits/vegetables different amongst the groups at baseline and at follow-up time periods? This is an important variable to assess given it is a significant contributor to blood/plasma carotenoid levels.
The reviewer is right, but although dietary history was collected, it was in general terms, and we do not have quantitative information on consumption of fruit and vegetables by the participants. However, the participants were asked to adhere to their usual diet; thus, the basal lycopene level can give partial information on differences in consumption and differences in bioavailability among them. The basal lycopene levels were added to Table 1 and are described in lines 176–178. The study limitations including the lack of detailed dietary information were added to the Discussion (lines 281–293).
- CBC data was collected from participants at enrollment and followup, no note in made in the results as to whether this changed or remained constant, please comment.
Corrected – CBC was added to the list of blood tests that did not change (line 170).
Results:
- Table 1 should be updated to include demographic and clinical data associated with risk of hypertension BMI, race, ethnicity, HDL, LDL, smoking status, alcohol consumption and TG
Baseline data of lycopene, TG, HDL, and LDL were added to Table 1. Although BMI is very important in BP studies, these data were not found in the final report of the study and cannot be added. This limitation is referred to in the Discussion (lines 290–291). Data for race, ethnicity, smoking status, and alcohol consumption were not collected.
- Data from Table 2 and Figure 2 suggest the TNC dose with the greatest benefit on blood pressure is 15 mg, however data in Figure 3 would suggest that serum increases in lycopene do not differ greatly between the different treatment doses. This is further supported by data from Table 6 which shows that lycopene dose not increase in serum in a dose-dependent nature. Rather, phytofluene and phytoene increase in a dose-dependent nature - is there any evidence from the authors (or others) that these compounds may be having a beneficial effect on blood pressure?
We agree with the reviewer on the possible role of phytofluene and phytoene (and maybe other components of TNC) in the reduction of BP. This can also explain the low effect of synthetic lycopene, which does not contain phytoene and phytofluene. The reviewer’s suggestion was added to the Discussion, and the last two paragraphs of the original Discussion were rewritten to include it (lines 339–355).
- Can the authors provide insight into the proposed mechanism of action of THC and reductions in blood pressure within the discussion - given they comment it is not due to reductions in ROS production.
We did not write that the effect is not due to “reductions in ROS production”. We only cited a paper [Devaraj et al. (ref 18)] showing that synthetic lycopene, which was not effective in the current study, does not reduce lipid peroxidation. We actually believe that part of the BP lowering effect is due to reduction in ROS production, but since we did not measure ROS levels, we did not suggest it in the original manuscript. In the revised manuscript, according to the reviewer’s recommendation, we suggest this mechanism in the Discussion section (lines 356–364), based on a paper describing the effect of lycopene and beta-carotene on ROS and NO in endothelial cells (Di Tomo et al. 2012).
- The population used in this study has SBP/DBP values considered an "at risk" population, and THC reduced blood pressure to the normal range. Clinically, is the THC product anticipated to be used in conjunction with optimal medical therapy for the lowering of hypertension, or as a stand alone product? Or a preventative measure?
The possible pathways to use TNC in HT patients, suggested by the reviewer, were added to the Conclusions (lines 372–376).
Table 6
p values should not be reported as 0.000, unless it truly is, these should be updated to P<0.000x.< span="">
Corrected in Table 6, which is now Table 7.
Reviewer 2 Report
This is study examines the dose-response of TNC vs placebo vs synthetic lycopene on blood pressure in people with stage 1 and 2 HT. The study extends to include an open-label component for 4 months in some subjects.
The paper also includes a separate study examining different doses of TNC in younger individuals and over 4 weeks.
The study corresponds to clinical trials ID NCT00637858; however, this ID corresponds to a 12-week study with Lyco-mato, with 130 enrolled. Is this the correct study identifier for the investigations being reported in this submission? If not, the authors need to provide the correct ID. If yes, the authors need to provide how the reported studies here relate to the details of the study described in NCT00637858.
BP is the primary endpoint: How was the sample size determined (what specific information was used/expected for effect size and variance between and within treatments and subjects to justify the samples per group? This should be included in the statistical section. How many blood pressure instruments were used and how were they calibrated? It appears a study nurse measured all blood pressures. The authors indicate that BP measurements were taken at the same hour of the morning after abstinence from food and caffeine for a minimum of 30 min. However, this needs to be looked at more clearly. Significant increases in blood pressure occur 1 h postprandially and continue to change (increase and decrease) over the next 2-3 h. This could add significant variance in the final data depending if a subject was measured at different times since they last ate (ie., 1 h vs 2 for example). In addition, what the subject consumed could impact the magnitude of change postprandially.
The authors should consider including a table of dietary intake before blood pressure measurements, including time since measurement. Further, the analysis should take this into account in the model.
Caffeine absorption peaks over 1-3 h, so depending on the variance in caffeine intake, metabolic variance in subjects (genotype for caffeine metabolism), this too can impact results. The authors need to address this/these issues.
15 mg synthetic lycopene and placebo capsules containing soybean oil. Was the lycopene in TNC and in synthetic form matched beyond amount, ie., in the same configuration (cis or trans lycopene or both)? What was the ratio of cis:trans and were their shorter lycopene fragments in the formulations? How would the authors expect this to impact plasma and outcome results?
It is unclear why the authors included another study of for bioavailability. These subjects were Younger subjects (20-40 y), which could impact bioavailability, and studied for 4 weeks only. Also, different doses were studied and no synthetic lycopene was studied. Outcomes included lycopene plus other carotenoids. These type of data are more useful from subjects in the BP study. Since plasma lycopene results were reported in the BP, why not other carotenoids?
The younger person, 4 weeks, bioavailability data should be written up as a separate study or justified better for why it should be included in this report.
As a matter of form… inconsistency in study purpose needs to be addressed.
Note in the introduction, authors write that “the major aims of the current work was to compare the effect on BP of pure synthetic lycopene to a proprietary tomato extract. To complement our previous results showing the improvement of BP by natural antioxidants from tomato extract in pre-HT patients [6,9], we tested its BP-lowering effects 60 in Stage 1 and 2 HT patients, attempting to establish a dose dependency and to show bioavailability of the major carotenoids present in tomato extract.”
But then…. In the Discussion write:
“The aim of the current study was to perform a dose-response analysis and uncover the optimal effective dose of a proprietary Tomato Nutrient Complex supplement in maintaining blood pressure within a normal range among untreated HT individuals. Results of this ……..” The true purpose of the study needs to consistent throughout and the statistical analysis conducted to test the primary hypothesis and subsequently the secondary hypotheses and associated endpoints.
Author Response
Review 2
This is study examines the dose-response of TNC vs placebo vs synthetic lycopene on blood pressure in people with stage 1 and 2 HT. The study extends to include an open-label component for 4 months in some subjects.
The paper also includes a separate study examining different doses of TNC in younger individuals and over 4 weeks.
The study corresponds to clinical trials ID NCT00637858; however, this ID corresponds to a 12-week study with Lyco-mato, with 130 enrolled. Is this the correct study identifier for the investigations being reported in this submission? If not, the authors need to provide the correct ID. If yes, the authors need to provide how the reported studies here relate to the details of the study described in NCT00637858.
NCT0063785 is the correct study. Although it is described as “overall 12 weeks”, it is written in the description of the various treatments: “Daily [compound and dose] with lunch for 8 weeks (After 4 weeks of placebo run in)”. Thus, the 12 weeks refer to the sum of the run in and treatment periods. Information about the interim analysis, which led to ending the study before recruitment of 130 patients, was added in a separate paragraph in section 2.1 (lines 65–68).
BP is the primary endpoint: How was the sample size determined (what specific information was used/expected for effect size and variance between and within treatments and subjects to justify the samples per group? This should be included in the statistical section. How many blood pressure instruments were used and how were they calibrated?
Data used in sample size calculations were added in the Statistics section 2.5 (lines 145–148).
A sentence was added to specify that BP measurements were done with one new and calibrated instrument, which was used only for this study (lines 92–93).
It appears a study nurse measured all blood pressures. The authors indicate that BP measurements were taken at the same hour of the morning after abstinence from food and caffeine for a minimum of 30 min. However, this needs to be looked at more clearly. Significant increases in blood pressure occur 1 h postprandially and continue to change (increase and decrease) over the next 2-3 h. This could add significant variance in the final data depending if a subject was measured at different times since they last ate (ie., 1 h vs 2 for example). In addition, what the subject consumed could impact the magnitude of change postprandially.
The authors should consider including a table of dietary intake before blood pressure measurements, including time since measurement. Further, the analysis should take this into account in the model.
Caffeine absorption peaks over 1-3 h, so depending on the variance in caffeine intake, metabolic variance in subjects (genotype for caffeine metabolism), this too can impact results. The authors need to address this/these issues.
The reviewer is right that the type of food and the exact time it was taken can affect BP measurement. While 30 min was the minimum time between food intake and BP measurement, the exact time and the type of food/beverages taken were not recorded. It is clear that with an improved design, the results might be better than those achieved in the current study. To address this and similar issues, a paragraph was added to describe the study limitations (lines 281–293).
15 mg synthetic lycopene and placebo capsules containing soybean oil. Was the lycopene in TNC and in synthetic form matched beyond amount, ie., in the same configuration (cis or trans lycopene or both)? What was the ratio of cis:trans and were their shorter lycopene fragments in the formulations? How would the authors expect this to impact plasma and outcome results?
The information about % cis isomers of TNC (8–10%) and synthetic lycopene (18–20%) was added to the formulation description (lines 83, 85). We do not expect these differences in the cis:trans ratio to affect the study outcome because we showed (Walfisch et al. 2003) that this ratio in the supplemented lycopene does not affect that in plasma and tissues, which does not change after supplementation in spite of a large increase in plasma concentration. We added this information and the reference to the Discussion (lines 323–326). We do not have information about shorter lycopene fragments in the formulations used in the study.
It is unclear why the authors included another study of for bioavailability. These subjects were Younger subjects (20-40 y), which could impact bioavailability, and studied for 4 weeks only. Also, different doses were studied and no synthetic lycopene was studied. Outcomes included lycopene plus other carotenoids. These type of data are more useful from subjects in the BP study. Since plasma lycopene results were reported in the BP, why not other carotenoids?
The younger person, 4 weeks, bioavailability data should be written up as a separate study or justified better for why it should be included in this report.
It is clear that carotenoid data from subjects in the BP study are more useful. Unfortunately, as was written in the Results section (lines 225–226), there was not sufficient information from the BP study to get valid results; thus, the information from a somewhat similar study was added. Although that study was not parallel to the BP study in terms of population and design, the information is still relevant and important, as suggested by Reviewer 1, who asked us to discuss the higher increase in phytofluene and phytoene concentrations as compared to that of lycopene. Although the bioavailability study was done with a different population, the information achieved about differences in the dose response of the three major carotenoids can help in understanding the results of the BP study. This message was added to the Discussion (lines 340–343).
As a matter of form… inconsistency in study purpose needs to be addressed.
Note in the introduction, authors write that “the major aims of the current work was to compare the effect on BP of pure synthetic lycopene to a proprietary tomato extract. To complement our previous results showing the improvement of BP by natural antioxidants from tomato extract in pre-HT patients [6,9], we tested its BP-lowering effects 60 in Stage 1 and 2 HT patients, attempting to establish a dose dependency and to show bioavailability of the major carotenoids present in tomato extract.”
But then…. In the Discussion write:
“The aim of the current study was to perform a dose-response analysis and uncover the optimal effective dose of a proprietary Tomato Nutrient Complex supplement in maintaining blood pressure within a normal range among untreated HT individuals. Results of this ……..” The true purpose of the study needs to consistent throughout and the statistical analysis conducted to test the primary hypothesis and subsequently the secondary hypotheses and associated endpoints.
Regrettably, we did not relate to synthetic lycopene when describing the aims in the discussion. To correct this, we added the following sentences in lines 278–281: “….and to compare the effect of TNC to that of synthetic lycopene. The bioavailability of the major tomato carotenoids was studied in a separate group of volunteers in order to gain insight into their relative contribution to the BP lowering effect of TNC.”
Reviewer 3 Report
Karen Li
Assistant editor
Nutrients
Ref Manuscript ID: nutrients-464117
Dear Dr Li,
Thank you for your invitation to review the manuscript by Wolak etal entitled " Effect of Tomato Nutrient Complex on blood pressure: A double blind, randomized dose-response study" submitted to Nutrients.
I read this paper with interest. The authors present the results of an RCT in an important topic. Overall, the study is well written.
However, some areas would benefit from clarification and further detail. Below I made recommendations for improvement before this manuscript is accepted for publication.
I strongly recommend the authors adopt the CONSORT guidelines (http://www.consort-statement.org/) to report their manuscript.
Abstract:
The abstract would benefit of information on actual results related to BP changes.
Methods section:
Further details of the randomisation process (allocation, concealment, and implementation) are currently lacking.
The authors should provide a brief description of sample size calculations.
The authors state that blood samples were obtained to assess a number of biomarkers, but results are not shown. Please state the reason why these are not reported.
Results section:
Section 3.4. The authors state that the large variability in serum lycopene and the small number of tested samples conditioned lack of statically significant differences. However, no information is provided as to how many samples were analysed and why. In the method section it is stated that plasma lycopene is to be analysed (presumably in all samples). It would be useful to readers if the authors provide further details of plasma concentrations observed (eg mean, standard deviation) rather than just changes as shown in figure 3 (This figure does not indicate the units of the changes shown).
Table 6 and figure 4 would benefit of clarifying the units used uM/?
Discussion
A section acknowledging the potential limitations associated with this study is currently lacking.
Although the researchers state that participants were encouraged to maintain their dietary habits, no evidence seems available to confirm this, Therefore the possibility that the changes in BP observed in this study could be attributed to potential changes in body weight, dietary changes, or physical activity.
Author Response
Review 3
Thank you for your invitation to review the manuscript by Wolak etal entitled " Effect of Tomato Nutrient Complex on blood pressure: A double blind, randomized dose-response study" submitted to Nutrients.
I read this paper with interest. The authors present the results of an RCT in an important topic. Overall, the study is well written.
However, some areas would benefit from clarification and further detail. Below I made recommendations for improvement before this manuscript is accepted for publication.
I strongly recommend the authors adopt the CONSORT guidelines (http://www.consort-statement.org/) to report their manuscript.
We now understand that the CONSORT guidelines are the correct way to describe clinical studies; however, this is a time-consuming request that will require major changes to the manuscript – a task we cannot accomplish in the time allocated for the revision. If absolutely required, we can later add the CONSORT checklist as supplementary information.
Abstract:
The abstract would benefit of information on actual results related to BP changes.
The limit on the abstract length does not allow inclusion of detailed information. We will include the actual results if the editors will allow a longer abstract.
Methods section:
Further details of the randomization process (allocation, concealment, and implementation) are currently lacking.
Information was added (lines 102-108)
The authors should provide a brief description of sample size calculations. Added (lines 146-149)
The authors state that blood samples were obtained to assess a number of biomarkers, but results are not shown. Please state the reason why these are not reported.
We wrote in the manuscript (lines 169–171): “nor were there any significant changes in glucose, urea, creatinine, uric acid, sodium, potassium, chloride, cholesterol, triglycerides, AST, ALT, ALP, LDH, HDL, or LDL levels.”). Since there were no changes in the biomarkers, we decided not to add additional tables with these results, as they will not add important information.
Results section:
Section 3.4. The authors state that the large variability in serum lycopene and the small number of tested samples conditioned lack of statically significant differences. However, no information is provided as to how many samples were analysed and why. In the method section it is stated that plasma lycopene is to be analysed (presumably in all samples). It would be useful to readers if the authors provide further details of plasma concentrations observed (eg mean, standard deviation) rather than just changes as shown in figure 3 (This figure does not indicate the units of the changes shown).
A new table, Table 6, which includes the requested information, was added before Figure 3. The units which were missing in the figure were added to this, and to all other figures. The original Table 6 (now 7) did include the units (µM) at the top left [Carotenoid plasma concentration (µM)] but not at each column.
Table 6 and figure 4 would benefit of clarifying the units used uM/?
Discussion
A section acknowledging the potential limitations associated with this study is currently lacking.
Although the researchers state that participants were encouraged to maintain their dietary habits, no evidence seems available to confirm this, Therefore the possibility that the changes in BP observed in this study could be attributed to potential changes in body weight, dietary changes, or physical activity.
The lack of dietary information and other study limitations were added in the Discussion (lines 285–291)